Mitochondrial genomics of human pathogenic parasite Leishmania (Viannia) panamensis

Urrea Daniel Alfonso 1 2
Triana-Chavez Omar 2
Alzate Juan F. jfernando.alzate@udea.edu.co 3
1 Laboratorio de Investigaciones en Parasitología Tropical (LIPT), Departamento de Biología, Facultad de Ciencias, Universidad del Tolima , Ibague , Tolima , Colombia
2 Grupo Biología y Control de Enfermedades Infecciosas (BCEI), Universidad de Antioquia , Medellín , Antioquia , Colombia
3 Centro Nacional de Secuenciación Genómica -CNSG, Sede de Investigación Universitaria -SIU. Grupo de Parasitología, Facultad de Medicina, Universidad de Antioquia , Medellin , Antioquia , Colombia
Uversky Vladimir
Electronic publication date: 2019 Jul 2
Publication date: 2019
Volume: 7
Electronic Location ID: e7235
Received 2019 Feb 25; Accepted 2019 Jun 1
Copyright: ©2019 Urrea et al.
Copyright year: 2019
Copyright holder: Urrea et al.
License: This is an open access article distributed under the terms of the Creative Commons Attribution License, which permits unrestricted use, distribution, reproduction and adaptation in any medium and for any purpose provided that it is properly attributed. For attribution, the original author(s), title, publication source (PeerJ) and either DOI or URL of the article must be cited.
License URL: https://creativecommons.org/licenses/by/4.0/

Keywords: Leishmania panamensis, Genomics, Mitogenome, Mitochondria, Comparative genomics, Ngs

Funding: Universidad de Antioquia UdeA, and by a Ph.D. Studentship funded by Colciencias 11551929249 This work was supported by Colciencias grant 11551929249, Universidad de Antioquia UdeA, and by a Ph.D. Studentship funded by Colciencias. The funders had no role in study design, data collection and analysis, decision to publish, or preparation of the manuscript.

==============================
Background

The human parasite Leishmania (V.) panamensis is one of the pathogenic species responsible for cutaneous leishmaniasis in Central and South America. Despite its importance in molecular parasitology, its mitochondrial genome, divided into minicircles and maxicircles, haven’t been described so far.

Methods

Using NGS-based sequencing (454 and ILLUMINA), and combining de novo genome assembly and mapping strategies, we report the maxicircle kDNA annotated genome of L. (V.) panamensis, the first reference of this molecule for the subgenus Viannia. A comparative genomics approach is performed against other Leishmania and Trypanosoma species.

Results

The results show synteny of mitochondrial genes of L. (V.) panamensis with other kinetoplastids. It was also possible to identify nucleotide variants within the coding regions of the maxicircle, shared among some of them and others specific to each strain. Furthermore, we compared the minicircles kDNA sequences of two strains and the results show that the conserved and divergent regions of the minicircles exhibit strain-specific associations.

Introduction

The mitochondrial DNA of kinetoplastid parasites (kDNA) can represent as much as 20–25% of the total amount of DNA per cell (De Souza, 2002) and consists of two classes of circular molecules that form a dense concatenated network. Maxicircles, with 20–40 copies per cell, around 20–40 kb in size, are considered homogeneous in sequence within each parasite (Maslov, Kolesnikov & Zaitseva, 1984; Shapiro & Englund, 1995; Lukes et al., 2002; Shlomai, 2004), and minicircles with several thousand copies per network, which are heterogeneous in sequence and size (Simpson, 1987).

Maxicircles are functional DNA molecules equivalent to the mitochondrial genome of other eukaryotes, contain rRNA genes (12S and 9S), and coding genes for proteins mainly involved in the oxidative phosphorylation pathway (Simpson, 1987). The genome of the maxicircle contains 18 protein-coding genes including ribosomal protein S12 (RPS12), ATP6, cytochrome B (CyB); subunits I, II and III of cytochrome c oxidase (COI, COII, COIII), and subunits 1, 2, 3, 4, 5, 7, 8 and 9 of the NADH dehydrogenase (ND1, ND2, ND3, ND4, ND5, ND7, ND8 and ND9). The remaining genes are annotated as MURF2, MURF5, G3, and G4 (Lin et al., 2015; Bhat et al., 1990; Simpson, 1986; De la Cruz, Neckelmann & Simpson, 1984). The synteny of these genes is highly conserved among kinetoplastids of the genera Trypanosoma, Leishmania, and Leptomonas (Westenberger et al., 2006; Yatawara et al., 2008). Twelve of these genes are cryptogenes, whose transcripts need to be edited in order to be properly translated. Some transcripts lack codons that indicate the start of translation and others require editing inside the coding regions. Some of them require minor editing events with few uridines involved, while others require the addition of hundreds (pan-editing). In several cases, the mRNA remodeling can affect more than 50% of the final length of the molecule. However, the length of the edition varies between species (Simpson et al., 2000; Shaw et al., 1988; Seiwert, 1995).

Minicircles form a concatenated network and are reported to be heterogeneous in sequence within each cell and can vary among strains. Each minicircle encodes one guide RNA - gRNA -, which will participate in the editing process of one of the maxicircle cryptogenes. These gRNAs interact with the mRNA through hybridization between the 5′ end of the gRNA and the 3′ end of its target mRNA to orient the precise location and number of insertions/deletions of uridine residues in the pre-mRNA for maturation. This RNA editing process creates start codons, corrects frameshifts and often adds fractions of coding sequences to create functional ORFs from cryptogene transcripts. Although most of the gRNAs can be assigned to one specific cryptogene, some of them can’t be related to any maxicircle gene and are considered “orphans”. Minicircles consist of a conserved region (CR), which include three conserved sequence blocks (CSBs) followed by regions rich in uridines that encode the small gRNAs, which present a high variability and are known as the divergent region (DR). Avila & Simpson (1995), Blum, Bakalara & Simpson (1990), Cruz-Reyes & Sollner-Webb (1996), Stuart et al. (2005), Simpson et al. (2015).

Understanding the composition and function of kDNA is essential for the adequate comprehension of the biology of the host-parasite relationship. The mitochondrial genomes have been described in a complete or partial manner in some kinetoplastid species such as Leishmania tarentolae (Simpson et al., 2015), L. major (Yatawara et al., 2008), L. donovani (Nebohacova et al., 2009), L. amazonensis (Maslov, 2010), L. lewisi (Lin et al., 2015), Trypanosoma brucei (Simpson, 1987), T. cruzi (Westenberger et al., 2006; Ruvalcaba-Trejo & Sturm, 2011), T. copemani (Botero et al., 2018), Crithidia fasciculata (Sloof et al., 1987), Phytomonas serpens (Maslov, Nawathean & Scheel, 1999), among others. Leishmania (Viannia) panamensis is an American unicellular parasite belongs to the Trypanosomatidae family (Banuls, Hide & Prugnolle, 2007), causing American cutaneous leishmaniasis, the most common clinical form of the disease in Central and South America including Colombia (Alvar et al., 2012), where L. (V) panamensis is responsible for at least half of the cases (Corredor et al., 1990; Munoz & Davies, 2006; Ovalle et al., 2006). Previously, the genome of the species was reported (Llanes et al., 2015) and recently the genomes of four Colombian strains with different levels of virulence were analyzed but no information regarding its whole mitochondrial genome has been published so far (Urrea et al., 2018). Here we report the assembly, annotation, and comparison of the maxicircle of L. (V) panamensis with other Trypanosomatid maxicircles previously reported. Additionally, the NGS-generated data allowed the study of the minicircles of this species and its comparison with previously reported results using PCR-amplification and capillary sequencing (Brewster & Barker, 2002). The interspecific and intraspecific variation is discussed.

Materials & Methods

Sequencing data and de novo assembly of the maxicircle

NGS DNA-seq data of the Colombian strain of L. (V.) panamensis UA946 is already available at the SRA database under the accession number SRP154327 and was download for de novo assembly purposes. Reads of two 454 sequencing experiments were downloaded and used, one shotgun experiment and one 8 kb 454 mate-paired experiment (MP). Additionally, the reads of one Illumina HiSeq shotgun DNA-seq PE (paired end reads 100 bases) experiment was downloaded. Initially, using the assembler Newbler v2.9 (Margulies et al., 2005), the 454 data (shotgun and 8 Kb MP reads) were de novo assembled using default settings. One scaffold of 16.335 bases and 4 gaps (3,671 “Ns”), which includes 5 contigs, was assembled and identified as carrying most of the maxicircle information using BLASTN v2.2.31 comparisons (Fig. 1A). To improve the maxicircle model, HiSeq shotgun reads were extended using the program FLASH v1.2.11 and mapped to the maxicircle model (Fig. 1B). Matching reads were extracted and mixed with the 454 shotgun reads that were already identified as of maxicircle origin. Again, a de novo genome assembly was carried out with Newbler v2.9 and the obtained contigs were used to fill the gaps of the first scaffold obtained with the 454 MP data. Leishmania tarentolae kDNA reference maxicircle complete sequence (accession M10126), L. donovani (accession FJ416603), L. major (accession EU140338) and L. amazonensis (accession HM439238) partial sequences were used to assist and support the scaffolding, filling and validation process (Fig. 1C). To validate the molecule obtained, PCR primers were designed at the flanking sites of the gaps, conflictive regions and ends of the molecule (File S1, Table S1, Fig. 1A). PCR products were amplified and capillary sequenced in MACROGEN (Seoul, Korea). High-quality chromatograms of the PCR products were used to manually curate the joining edges in ARTEMIS v16.0.0 (Carver et al., 2008). Finally, to improve the quality of the generated model, iCORN v0.97 (Otto et al., 2010) was used to correct errors in consensus sequences including homopolymer errors from pyrosequencing, by iteratively mapping the whole dataset of the Illumina HiSeq 100 bp paired-end reads. The mitochondrial genes were annotated through alignment between sequences of reference proteins and genes reported in the NCBI database. The L. (V.) panamensis UA946 maxicircle molecule was submitted to the GenBank under accession number MK570510.

Figure 1 Integrative genomics viewer (IGV) visualization of L. (V) panamensis kDNA maxicircle assembled.

(A) Initial scaffold assembled by 454 Mate Pair and shotgun reads. The arrows indicate the alignment sites of the primers designed to complete the molecule. (B) Mitochondrial Illumina reads extended to close gaps. (C) Raw Illumina read depth on the reassembled maxicircle without gaps. (D) Normalized read depth of final molecule.

kDNA minicircles analysis

Based on the results of the first de novo genome assembly described above of the shotgun 454 data, a search of kDNA minicircles was carried out. To do so, the complete contig dataset was compared with a local database built with previously reported kinetoplastid minicircles using BLASTN v2.2.31, option “blastn-short”. Additionally, we performed searches with query sequences of the three conserved sequence blocks of minicircles (CSBs), CSB1 (5′-GAACGCCCCT-3′), CSB2 (5′-GCACGGGG-3′) and CSB3 (5′-ATGTGGTTGGGG-3′) present in the same order and with similar spacing in all kinetoplast species (Ray, 1989; Yurchenko et al., 1999) and a 12-mer sub-genus Viannia specific sequence (5′-TAATTGTGCACGGGGA-3′) (Fernandes et al., 1996).

Comparison with other kinetoplastids

Synteny and nucleotide identity analysis with other kinetoplastid maxicircles were done using MAUVE version snapshot_2015_02_25 (Darling et al., 2004) and MUMmer v4.x (Delcher et al., 2002) programs, through alignments among the whole or partial maxicircle genomes previously reported. To evaluate the identity and editing patterns, the sequences of the coding regions comprising pre-edited genes also were aligned with their homologs in L. tarentolae (accession M10126), L. donovani (accession FJ416603), L. major (accession EU140338), L. amazonensis (accession HM439238), L. (V.) braziliensis (Ramirez, Puerta & Requena, 2011), T. brucei (accession M94286), T. cruzi (accession DQ343646) and T. rangeli (accession KJ803830).

To evaluate the intraspecific variability and the possible encryption variations (significant changes in the numbers of nucleotides added during the editing process) of mitochondrial genes, three more strains of L. (V.) panamensis were studied by a mapping approach using previously generated Illumina HiSeq data (100 bp PE reads, 350 bp insert size) (accession PRJNA481617). The reads from the SRA accessions PRJNA165959 (L. (V.) panamensis strain L13), accession PRJNA267749 (L. (V.) panamensis strain WR120) and accession PRJNA235344 (L. (V.) panamensis strain PSC-1) were also downloaded and mapped using BOWTIE2 v2.2.9 with default settings and SNVs (Single Nucleotide Variations) and indels calling was performed using NGSEP pipeline v3.1.0 (Duitama et al., 2014).

Regarding the DNA minicircles, in order to identify variations in L. (V.) panamensis molecules isolated in different time periods and geographical origin, the assembled L. (V.) panamensis DNA minicircles were compared with those previously reported for the species in 2002 by Brewster and Barker in the Nucleotide database (accessions: AF118454 –AF118474). The Conserved Region (CR) and the Divergent Region (DR) was evaluated separately. The CR were evaluated by a neighbor-joining tree and the DR by analysis of the composition of di, tri, tetra, penta and hexanucleotides determined by compseq tool, emboss package v6.6.0.0 (Rice, Longden & Bleasby, 2000), and the differences evaluated by multivariate statistical analysis through profile analysis for one sample Hotelling’s T-Square in R version 3.5.1 environment (R Core Team, 2018).

Results

Using only 454 shotgun and Mate Pair data, it was possible to assemble de novo a 16 kb scaffold that was confirmed to carry maxicircle genetic information. This scaffold, with an average depth of 7X, was built with five contigs and harbored four gaps. In order to improve this maxicircle molecule model, additional ILLUMINA HiSeq shotgun data was used in combination with the 454 shotgun reads to generate a new de novo assembly. This new assembly yielded 25 contigs of maxicircle origin that summed 19.957 bases. These additional contigs were used to fill the gaps using as reference the maxicircle genome of L. tarentolae. Contig joining regions were validated with nine directed PCR reactions and subsequent capillary sequencing. These PCR reactions also allowed confirming the circularity of the molecule. The final L. (V.) panamensis maxicircle molecule was corrected using the whole Illumina HiSeq dataset with iCORN. In the end, the definitive reference model molecule assembled of the maxicircle of L. (V.) panamensis has 19.393 bases in length, without harboring ambiguous bases. Mapping the complete read dataset, the maxicircle model showed a full coverage of the molecule and an average sequencing depth of 782X. It is important to point out that the repetitive AT-rich region that spans between the ND5 and 12S genes is complicated to assemble and can be longer since the mapping analysis showed a significant peak of reads above coordinate 19.000 and below coordinate 500 (Figs. 1C, 2, close to coordinate 0). However, based on normalized sequencing depth analysis it was possible to estimate that the total size of the molecule might be at approximately 22,800 bp. (Fig. 1D).

Figure 2 Graphical map of the maxicircle kDNA L. panamensis genome.

From outside to inside. The black line represents DNA molecule with its nucleotide coordinates. The colored rectangles represent the coding sequences. The lilac rectangles depict ribosomal RNAs genes. The blue histogram represents the read coverage of maxicircle assembled model. In the center, bars represent detected SNVs compared to L. (V.) panamensis UA946: black, L13 strain; light red, UA1114 strain; orange, UA1511 strain; purple WR120 strain; blue, PSC-1 strain; green, UA140 strain; resequencing of UA946 strain.

The annotation was carried out based on the alignment with reference sequences of proteins and genes of T. brucei, L. tarentolae, L. major, L. donovani, L. amazonensis and L. (V.) braziliensis (Ramirez, Puerta & Requena, 2011). The genetic code used was the one available for Mold, Protozoan, Coelenterate Mitochondrial from the ARTEMIS tool (Carver et al., 2008). The resulting molecule contains two ribosomal genes and 18 structural genes in perfect synteny, compared with other kinetoplastids, in the following order (ND8, ND9, MURF5, ND7, COIII, CyB, ATPase6, ND2, G3, ND1, COIII, MURF2, COI, G4, ND4, ND3, RPS12 and ND5) (Fig. 2).

Global alignment of the L. (V.) panamensis maxicircle DNA sequence with its homolog references of L. tarentolae and L. donovani showed nearly 86% nucleotide identity; while, as expected, the nucleotide identity with Trypanosoma species drop to 55%, 53%, and 52%, in the case of T. brucei, T. cruzi and T. rangeli, respectively (Fig. 3). The same situation was found when a synteny analysis was done (Fig. 3B).

Figure 3 Comparative analysis of the maxicircles kDNA assembled to date.

Global alignment of the maxicircle of L. panamensis against different maxicircles of trypanosomatids reported. (A) L. tarentolae. (B) L. donovani. (C) T. brucei. (D) T. cruzi. (E) T. rangeli. The values correspond to the percentage of nucleotide identity (MUMmer program). (F) Dendrogram derived from the similarity in synteny between the maxicircles analyzed (Mauve program).

The coding sequences analysis, comprising pre-edited genes, with its respective homologs in L. (V.) braziliensis, L. major, L. donovani, L. mexicana, T. brucei, T. cruzi and T. rangeli; showed the highest nucleotide identity (97.7%) among the mitochondrial genes of L. (V.) panamensis and L. (V.) braziliensis (Nocua et al., 2011). This result is in agreement with the reported taxonomic status of the isolate UA946 of L. (V.) panamensis and the phylogenetic relationship of subgenus Viannia (Table 1). Then, the coding regions of the L. (V.) panamensis maxicircle showed to be highly conserved among its homologs in L. donovani (82,4%), L. tarentolae (83%), L. amazonensis (81,5%) and L. major (82,7%). The non-coding regions of the maxicircle showed significantly lower nucleotide conservation between the compared species.

Table 1 Nucleotide identity among mitochondrial genes of L. panamensis and Leishmania spp. and Trypanosoma spp.

The intensity of the color corresponds to the nucleotide identity found.

L. (V.) panamensis	
	12S rRNA	9S rRNA	ND8	ND9	MURF5	ND7	COIII	Cyb	ATPase 6	ND2	ND1	COII	MURF2	COI	ND4	ND3	rps12	ND5	
L. (V.) braziliensis	98,9	99,5	96,5	97,3	98,7	98,2	98,6	98,6	.	.	97,3*	99,6	.	94,6*	98,4	98	93,7	97,1	
L. donovani	91,2	91,8	80,6	72,7	69,3	88,4	85,4	89,9	86,1	81	81,8	86,8	86,2	83,7	85,4	71,3	67,5	84,5	
L. tarentolae	86,4	92,8	79,8	74	69	88,3	86,7	90,8	87,6	84,5	79,7	88,2	85,6	83,8	84,5	65,3	#	84,7	
L. amazonensis	86,7	92,7	80	72,7	74,4	86,6	86	89,6	86,6	84,8	80,4	88,5	.	.	.	67,4	65,2	.	
L. major	84,1	91,7	79,8	68,3	74,2	85,2	84,7	88,9	84,4	83,7	81,2	86,3	.	.	.	.	.	.	
T. brucei	77,7	81,7	61,5	#	64,5	#	#	84,6	#	77,7	70,6	79,1	77	76	77,2	#	#	78,3	
T. cruzi	75,9	81,5	60	#	62,4	#	#	82,7	#	76,8	68,8	80,3	74,7	76,5	77,7	#	#	75,2	
T. rangeli	79,6	79,8	63,2	#	60	#	#	82,2	#	75,9	64,5	79,1	72,5	74,1	76,4	#	#	75,2	
Polarity	→	→	→	←	←	→	→	→	→	←	←	→	→	←	→	←	→	→	
Notes.

# Less than 60% identity.

* Partial sequence reported.

. No information available. B. Editing patterns found through the alignment of the maxicircle kDNA genome assembled and the promastigote and amastigote transcriptome assembled.

To evaluate the intraspecific variation in the L. (V.) panamensis maxicircle, the presence of SNVs was assessed using a mapping strategy of Illumina HiSeq reads of three strains of the same geographical origin and other L. (V.) panamensis reference strains. Fig. 2 depicts the localization of the SNVs found when the strains UA140, UA1114, UA1511, L13, WR120, and PSC-1 were compared with maxicircle model of UA946 strain. A higher accumulation of SNV was observed within the repetitive DR region that might be due to the mapping bias along the repetitive stretches. The coding regions of ND5 and rpl12 were the genes with more variations among all the compared strains. Indels in mitochondrial coding regions in the same strains also were evaluated. Eight genes showed indels in different positions in the genes ND3, ND8, ND9 and COIII in the Colombian strain L13 in comparison with the other strains. However, the MURF2, G4, RPS12, and ND5 genes showed indels alternative homozygous in different positions in all the strains analyzed (File S2). In general, the strain L13 was the one that presented more variations along the whole mitochondrial genome.

The minicircles analysis identified 21 different classes with an average size of 742 bp and an average GC content of 29.8% (File S3). Although no molecular tests were performed to verify the circular nature of the 21 minicircles, these results are in complete agreement with the minicircles of L. (V.) panamensis previously reported by Brewster and Barker in 2002 (File S1, Table S2). The multiple alignments of the 21 minicircles of L. (V.) panamensis previously reported and the 21 minicircles assembled here allowed to identify the 200 bp of the conserved region (CR), including the conserved sequence blocks CSBs and the divergent region (DR) with around of 550 bp in length (Fig. 4).

Figure 4 Comparative analysis of the minicircles of two strains of L. (V) panamensis sequenced in different time periods.

(A) Spectrogram representation of alignment of minicircle sequences assembled in the present work (top) and the minicircles reported by Brewster and Barker in 2002 (bottom). (B) Dendrogram derived from analyzing of the CR of minicircles kDNA analyzed. White circle, present work. Black circle, (Brewster & Barker, 2002). The values above the nodes correspond to bootstrap percentages derived from 1,000 replications from neighbor joining (NJ) analysis. Plot of nucleotide composition comparative from DR of minicircles kDNA analyzed. Up to 500 combinations of words used as variables were plotted with the results of profile analysis for one sample Hotelling’s T-Square. (C) Dinucleotide profile. (D) Trinucleotide profile. (E) Tetranucleotide profile. (F) Pentanucleotide profile. (G) Hexanucleotide profile.

To evaluate the similarity between the minicircles assembled in 2002 by Brewster and Barker, and the ones reported here, the differences between the CR (conserved region) and the DR (divergent region) were estimated. Independently of the low support values, the dendrogram based in the 200 bp of CR shows a strain-specific relationship trend in L. (V.) panamensis minicircles in the built nodes (Fig. 4B).

On the other hand, the lack of homology among the 42 DR from the minicircles aligned showed the need to make an additional analysis. Profile analysis is a multivariate technique to identify if two or more groups show a significantly different profile of variables. One of the applications of the test is to demonstrate that groups are not different (null hypothesis). To gain an insight into the relationship of the minicircles sequences, the following nucleotide composition were evaluated: 16 dinucleotides, 64 trinucleotides, 251 tetranucleotides, 842 pentanucleotides, and 2,227 hexanucleotides; for a total of 3,400 variables in the two analyzed groups, which represent the DR of the minicircles of the two analyzed strains (File S4). The profiles of the multivariate analysis allowed to reject the null hypothesis that the ratio of means of overall nucleotide composition of the divergent region (DR) of minicircles reported here and the ones published in 2002 are the same (T2 = 2118.45; F = 1058.91; df = 3, 398; p < 0.0001). However, the profiles analysis of the composition of di and trinucleotides show no difference between the two sets compared (Fig. 4C).

Discussion

In this report, we present the annotated sequence of the maxicircle of an infective strain of Leishmania (V.) panamensis, as well as its sequence comparisons with other reference mitochondrial genomes of Trypanosomatid parasites. The assembled 19,4 kb L. (V.) panamensis maxicircle is shorter than the 21 kb previously reported for L. tarentolae (de la Cruz et al. 1984) and the partial 20 kb reported for the maxicircle of L. donovani (Nebohacova et al., 2009). Similarly, it is shorter than previously reported for other kinetoplastids such as T. rangeli (25,3 kb) (Stoco et al., 2014), T. brucei (22,2 kb) (Stuart, 1979), T. cruzi strains Esmeraldo (22 kb) and CL Brenner (20 kb) (Westenberger et al., 2006), T. lewisi (20,6 kb) (Lin et al., 2015) and the partial genome of T. copemani maxicircle (19,2 kb) (Botero et al., 2018). However, all the reported coding regions for Leishmania maxicircle are present and maintain conserved synteny.

The main difference in size between the maxicircle genome model of L. (V.) panamensis presented here and other kinetoplastids is attributed to variability of the repetitive region and probably related to the collapse of the short read assembly on this long stretches of repetitive DNA (Westenberger et al., 2006; Franzen et al., 2012). Based on the normalized sequencing depth of the repetitive region, the size of the complete maxicircle was estimated at approximately 22,800 bp, closer to previous reports in other species. Hybridization assays with specific probes in L. (V) braziliensis, another species of Viannia subgenus, estimated the size of the molecule in 23 kb (Nocua et al., 2011).

The sequence identity of the mitochondrial pre-edited genes is significantly lower when compared with the Trypanosoma genus, probably reflecting different editing patterns that evolved independently in both genera (Feagin et al., 1988). The alignment analysis of pre-edited mitochondrial genes of Leishmania spp. and L. (V.) panamensis, with their respective expression products, showed three genes without evidence of the RNA editing process (Cyb, COI, ND1). The rest of the structural genes showed to need some degree of RNA editing. Previously, it was reported that the ND8 in promastigotes of L. (V.) braziliensis must be edited to produce a functional transcript (Ramirez, Puerta & Requena, 2011). In T. cruzi, the ND7 gene should be extensively edited to be functional and the role of editing has been suggested as necessary in the pathology of the parasite (Baptista et al., 2006). COIII requires editing at the 5′in L. tarentolae but is extensively edited in T. brucei and T. cruzi (Maslov et al., 1994). In general, different editing patterns have been found for the same genes in different kinetoplastids and different experimental conditions. Maslov (2010), analyzing one strain of L. amazonensis maintained axenic for long periods of time, found that all its cryptogenes were pan-edited. However, other studies have shown evidence of the co-existence of mitochondrial mRNA populations with a different degree of editing in Leishmania spp. Additionally, maxicircle heteroplasmy was confirmed in similar studies analyzing the same strain of L. major, in which changes in the predominance maxicircle can take place during the differentiation of promastigote to amastigote state under stress conditions (Flegontov & Kolesnikov, 2006; Flegontov et al., 2009). Previously, it was reported that the ND8 in promastigotes of L. (V.) braziliensis must be edited to produce a functional transcript. Similarly, indels variation of the same gene was found when compared with L. tarentolae, L. donovani and L. amazonensis (Ramirez, Puerta & Requena, 2011). Here, based on Illumina reads mapping, four genes (MURF2, G4, RPS12, and ND5) showed evidence of indels variability in the six strains analyzed. Ruvalcaba-Trejo & Sturm (2011), showed large numbers of indels at 5′ edited, extensively edited and not edited maxicircles genes in three strains of T. cruzi. Unfortunately, without evidence of posttranscriptional activity, it is not possible to infer the effect of these indels on the RNA edition. All these pieces of evidence, added to the results shown here, can suggest that some unknown properties of Leishmania can play a role in the maintenance and functioning of the RNA editing system and are important in the biology of the parasite (Simpson et al., 2015).

By comparing the intraspecific variability of the L. (V.) panamensis maxicircle through mapping analysis, the Colombian strain L13 (accession PRJNA165959) shows the highest variability of SNVs and Indels in comparison with the other strains evaluated (Fig. 2, File S2). Some strain-specific SNVs were found in all cases. Previously in T. cruzi, Westenberger et al. (2006) also reported frame shits and strain-specific SNVs in the coding regions. The repetitive AT-rich region, close to the coordinate 0 in Fig. 2, has the higher density of SNVs and indels, many of which can be artifacts due to the repetitive nature of the mapped reads in this region.

As originally reported for this species, we found 21 different classes of minicircles with almost equal lengths and percentages of GC (Brewster & Barker, 2002). A previous work published by Ceccarelli et al. (2017), based on phylogenetic analysis of the conserved region (CR), allowed to differentiate viscerotropic and cutaneous species causing Leishmaniasis. In the present work, evidence of differences in the conserved region (CR) of the minicircles between both strains of L. (V.) panamensis was found (Fig. 4B). Contrasting results have been obtained in the case of T. cruzi since it was not possible to detect differences in the conserved region (CR) between the minicircles of the Esmeraldo and CL Brener strains (Thomas et al., 2007). However, the two strains analyzed here have a lag of 16 years in terms of sequencing interval and technology used. In L. tarentolae, it was demonstrated that the frequencies of its 20–24 classes of minicircles change significantly due to continuous in vitro culture passage from one year to the next (Simpson et al., 2015).

Earlier, it was demonstrated species-specific profiles of dinucleotide frequencies in the Trypanosomatid minicircles (De Oliveira Ramos Pereira & Brandao, 2013). In the present work, we found differences in the composition of tetra, penta, and hexanucleotides of the divergent region (DR) of the minicircles of the two L. (V.) panamensis strains sequenced with a gap of time of 16 years. The nucleotide composition similarity between the minicircles of two strains isolated in different periods of time could suggest structural and functional stability in the minicircles. Rodrigues et al. (2013), demonstrated that the minicircles of isolates of L. (Viannia) spp., coming from patients with American cutaneous leishmaniasis with active lesions or scars of cured patients, show no differences in its dinucleotide profile composition. However, they found differences in the composition of tri, tetra, penta or hexanucleotides between the strains isolated from the two types of clinical conditions. Similarly, the sequence analysis of minicircles kDNA from 21 L. major strains showed correlations among the geographical origin and the clinical manifestations of cutaneous leishmaniasis (Oryan et al., 2013). kDNA minicircle sequences analysis of 37 L. donovani strains also yielded a different phylogenetic distribution but not associated with geographic origin or clinical symptoms (Jaber et al., 2018). Moreover, strain-specific gRNAs, have been found in different DTUs of T. cruzi (Thomas et al., 2007; Ortiz, Osorio & Solari, 2017).

Conclusions

The Leishmania Viannia subgenus mitochondrial genome present the same conservation pattern observed in other kinetoplastid species. Nonetheless, single nucleotide variations were observed within the coding and intergenic regions of the maxicircle. Some variants were identified as strain specific. Genes ND5 and rps12 accumulated the largest number of SNVs within the studied L. (V.) panamensis isolates. Minicircle molecules presented higher variation rates.

Supplemental Information

File S1 Primers and stats of minicircles

Click here for additional data file.

File S2 Indels calling of L. panamensis maxicircle

Click here for additional data file.

File S3 Maxicircle sequences in fasta format

Click here for additional data file.

File S4 Nucleotide composition of minicircles kDNA of L.(V.) panamensis

Click here for additional data file.

We thank Dr. Felipe Cabarcas for the information technology support.

Additional Information and Declarations

Competing Interests

Author Contributions

DNA Deposition

Data Availability

The authors declare there are no competing interests.

Daniel Alfonso Urrea and Juan F. Alzate conceived and designed the experiments, performed the experiments, analyzed the data, prepared figures and/or tables, authored or reviewed drafts of the paper, approved the final draft.

Omar Triana-Chavez contributed reagents/materials/analysis tools, authored or reviewed drafts of the paper, approved the final draft.

The following information was supplied regarding the deposition of DNA sequences:

Data is available at GenBank, accession number: MK570510.

The following information was supplied regarding data availability:

The raw data is available at the SRA database, accession numbers: SRP154327, PRJNA165959, PRJNA235344, PRJNA481617, PRJNA267749.

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
