# Peer review of "Mitochondrial genomics of human pathogenic parasite Leishmania (Viannia) panamensis"

_PeerJ, doi:10.7717/peerj.7235_

## Round 0.1 · original submission · Major Revisions

Please address critiques of the revisers and revise your manuscript accordingly

Reviewer 1 ·

Basic reporting

No comment

Experimental design

No comment

Validity of the findings

No comment

Additional comments

In this manuscript, the authors describe the assembly of the maxicircle coding region for Leishmania panamensis using NGS sequence reads. Additionally, the sequence of several minicircles was determined. Finally, comparisons with other previously reported sequences of maxicircles and minicircles are presented. The manuscript provides valuable information, but flaws in interpretation may exist. Moreover, the writing of the manuscript, mainly in the Results and Discussion sections, results unpleasant. The meaning of several sentences is really confusing. These deficiencies prevent to appreciate the consistency and relevance of this work. Therefore, the manuscript would benefit by an extensive editing, and this is a compulsory step before its evaluation for publication.

Major revisions.

1. In the abstract, at many places along the manuscript, and in figure 1, the authors claim that a complete maxicircle sequence is provided. However, the read coverage observed in the maxicircle divergent region (DR) is indicating that a collapse occurred during the assembly. This region is composed by low-complexity repeated sequences and complete sequencing of the maxicircle has not been attained yet for any Leishmania species (Flegontov et al. 2006. The Leishmania major maxicircle divergent region is variable in different isolates and cell types. Mol Biochem Parasitol 146: 173-179).
The authors should be cautious, avoiding to conclude that a complete maxicircle sequence has been assembled.

2. Lines 41-42. It is read: ‘Maxicircles, with 20-40 copies per cell, around 20-25 kb in size…”. However, this reviewer failed to find these data about the maxicircle size in the quoted references (Liu et al, 2005; Simpson et al., 1987). Looking for articles dealing with the direct determination of the maxicircle size, I found that sizes close to 40-kb have been estimated for this molecule in several trypanosomatids (Maslov DA et al. 1984. Mol Biochem Parasitol 12: 351-364).

3. Line 143. Could the authors explain the meaning “encryption variations”?

4. Lines 163-165. This sentence is confusing. The authors should use direct language. It seems that the authors want to indicate that combining NGS reads they assembled a contig of 19,957 nucleotides. If this is the message, please, rewrite the sentence.

5. Lines 167-168. Please, specify how the PCRs confirmed the “circularity of the molecule”. When responding to this question, keep in mind the points 1 (above) and 6 (below).

6. Lines 173-175. According to this sentence, the authors seem to be suggesting that the assembled molecule might be incomplete. This point must be clearly stated in the text, and figure 1 has to be modified accordingly to avoid confusion regarding whether or not a full assemble was attained.

7. Lines 189-190. Figure 2B does not seem to correspond to a “synteny analysis”, but a phylogenetic analysis based on sequence alignments.

8. Lines 194-195. The authors indicate that the L. panamensis maxicircle share high sequence identity with L. braziliensis maxicircle, and the percentages of identity between the different pairs of genes are shown in table 1. However, the authors are not citing the work in which the L. braziliensis maxicircle was reported (Nocua P et al. 2011. Universitas Scientiarum 16: 29-50).

9. Lines 206-207. The sentence is confusing: Which are the six L. panamensis strains that share the two SNVs located in the ATPase6 gene? My suggestion would be rewritten the entire paragraph.

10. Line 213. The authors wrote: “the 21 minicircles assembled here”. This information is relevant, but it is not shown. The authors should provide the sequence of the 21 minicircles as a supplementary file (the sequences in figure 3A are not readable). Moreover, it should be indicated whether they are or not complete molecules.

11. Line 221. What is the meaning of “nodes without polyphyly”? In figure 3B, please, include the Leishmania species for the minicircles with the AF identifiers.

12. Lines 221-230. This part of the manuscript results hardly understandable. The authors should clearly indicate the value of this analysis and the achieved conclusions.

13. Lines 290-291. What is the meaning of “the two strains analyzed here have a lag of 16 years in terms of sequencing interval”.

14. The discussion is too long and disorganized. My suggestion is to reduce it, putting in value the information reported in this work.

15. No conclusions are in the conclusion section. A conclusion is a concrete finding derived from the experimental data.

·

Basic reporting

no comment

Experimental design

no comment

Validity of the findings

no comment

Additional comments

The paper describes the complete maxicircle kDNA annotated genome of L (V) panamesis. The paper is very well written, with the appropiated refrences, the approach is sound and the conclusions clear.

Reviewer 3 ·

Basic reporting

In this work, the authors report the assembly and annotation of the complete maxicircle sequence of Leishmania V. panamensis, and compare it with published genomes of other related parasites.

Overall clear and unambiguous writing, background sufficiently covered except where noted in general comments for the author.

Experimental design

The aim of the study is well-defined and the methods used are appropriate.

Validity of the findings

Conclusions are well stated and supported by the results shown.

Additional comments

1) From the introduction it is not clear what motivated authors to study L.V. panamensis assembly in particular. Introduction should be revised to include this information.
2) On line 84, “..Here we report the sequencing, assembly, annotation, and ….”. This statement seems misleading and should be changed accordingly as authors are not reporting sequencing as part of this work rather working with sequences available through public repository as described in Methods.
3) On lines 118-119, “…The mitochondrial genes were annotated through alignment between proteins and genes reported in databases”. It is not clear which database authors are talking about here.
4) Authors should provide the version number of different programs described in Methods. For example, on line 159 “..Hotelling’s T-square in R environment” which R version was used?
5) Some of the abbreviations, such as PE on line 100, are used without first defining it.

---

## Round 0.2 · accepted · Accept

Since all critiques were adequately addressed and the manuscript was revised accordingly, this version of your manuscript is acceptable now.

Reviewer 1 ·

Basic reporting

no comment

Experimental design

no comment

Validity of the findings

no comment

Additional comments

After reading the revised manuscript and the authors’ responses, this reviewer considers that the authors have amended sufficiently the manuscript, addressing adequately previous concerns.